# At the Crossroads of Life and Death: The Proteins That Influence Cell Fate Decisions

**DOI:** 10.3390/cancers14112745

**Published:** 2022-05-31

**Authors:** Vinesh Dhokia, John A. Y. Moss, Salvador Macip, Joanna L. Fox

**Affiliations:** 1Mechanisms of Cancer and Ageing Laboratory, Department of Molecular and Cell Biology, University of Leicester, Leicester LE1 7RH, UK; vd82@leicester.ac.uk (V.D.); sm460@leicester.ac.uk (S.M.); 2Leicester Institute of Structural and Chemical Biology, Department of Molecular and Cell Biology, University of Leicester, Leicester LE1 7RH, UK; jaym1@leicester.ac.uk; 3FoodLab, Faculty of Health Sciences, Universitat Oberta de Catalunya, 08018 Barcelona, Spain

**Keywords:** senescence, p53, BCL-2, apoptosis

## Abstract

**Simple Summary:**

Cellular senescence and apoptosis were historically thought of as two distinct cell fate pathways. However, many of the proteins involved are integral to both pathways. In particular, the ability of p53 to regulate both senescence and apoptosis meant it was seen as the decisive factor in these decisions, yet questions remain about its ability to select on its own the most appropriate cell fate according to each situation. Therefore, cell fates are no longer considered fixed endpoints but dynamic states that can be shifted given the right combination of activation and/or inhibitions of cofactors.

**Abstract:**

When a cell is damaged, it must decide how to respond. As a consequence of a variety of stresses, cells can induce well-regulated programmes such as senescence, a persistent proliferative arrest that limits their replication. Alternatively, regulated programmed cell death can be induced to remove the irreversibly damaged cells in a controlled manner. These programmes are mainly triggered and controlled by the tumour suppressor protein p53 and its complex network of effectors, but how it decides between these wildly different responses is not fully understood. This review focuses on the key proteins involved both in the regulation and induction of apoptosis and senescence to examine the key events that determine cell fate following damage. Furthermore, we examine how the regulation and activity of these proteins are altered during the progression of many chronic diseases, including cancer.

## 1. Introduction: The Cellular Responses to Damage

Over time, cells are subject to a variety of external and endogenous stresses. Faced with these stresses, cells mount an appropriate response with two main end points—survival or death. Survival may mean permanent cell cycle arrest in the form of senescence or repair and continued proliferation after a temporary arrest [1]. However, if the damage is irreparable, cell death in a variety of forms such as apoptosis, necroptosis or ferroptosis leads to clearance of the damaged cell and prevents damage from spreading to the microenvironment [2]. The exact cellular response to damage depends on many factors, including the origin and intensity of the stress as well as cell type. The balance of these key proteins together ultimately determines cell fate (Figure 1). Cancer cells, in particular, experience enhanced internal and external stresses due to restricted nutrient availability in the microenvironment and the activation of oncogenes to supercharge metabolism [3]. This review particularly focuses on p53, which sits at the heart of cell fate decisions on whether to undergo senescence, apoptosis or non-apoptotic cell death and the implications of this decision for preventing cancer. We explore the effects of different stresses on cells and the different outcomes for the cell and outline the current evidence for the role of p53 in cell fate.

## 2. p53 as a Master Regulator of Damage Pathways

p53 is one of the most well-studied proteins and has a key role in DNA damage repair, cell cycle arrest, apoptosis and senescence. Its domain structure is critical to its ability to participate in the different molecular mechanisms [4]. The N-terminal tandem transcription activation domains (TADs) are required for its gene induction activity in response to DNA damage and oncogene activation [5]. Adjacent to the TADs is a proline-rich domain, which contributes to p53s transcriptional activity but is also essential for restricting cell growth [6]. Next, there is the DNA-binding core domain, which is responsible both for DNA binding and for tetramerisation of p53, and finally, the C-terminal domain (CTD), which is highly unstructured and the site of numerous post-translational modifications [7] (Figure 2).

In unstressed cells, p53 is maintained at low levels by the E3 ubiquitin-ligase MDM2. However, when stress is detected, p53 is phosphorylated, which prevents its binding to MDM2 and stabilises its levels [8]. Once stabilised, p53 is able to initiate its translational response to the stress through the upregulation of a whole host of proteins. p53 is involved in controlling the responses to DNA damage, oncogene activation and reactive oxygen species (ROS)-induced stress, among many others [9].

The DNA damage response (DDR) is a complex pathway that has evolved to detect DNA damage and facilitate its repair. In humans, two proteins central to DDR are ataxia telangiectasia mutated (ATM) and ataxia telangiectasia and RAD3 related (ATR). ATM is primarily involved in responding to double-strand breaks whilst ATR senses stalled or stressed DNA replication forks and is particularly important in detecting DNA damage in the S phase [10,11]. Once activated, ATM and ATR phosphorylate key proteins, including p53, MDM2, BRCA1 and the cell cycle checkpoint proteins CHK1 and CHK2 [12]. The inhibitory phosphorylations on CHK1 and CHK2 slow down or block cell cycle progression to allow time for DNA repair prior to replication and mitosis [13]. Moreover, phosphorylation of p53 on Ser-15 by either ATM or ATR prevents degradation by the ubiquitin ligase MDM2 [14] and induces a temporary cell cycle arrest [15]. If the DNA damage cannot be repaired, then the cell must commit to either senescence or cell death. Activation of p53 results in transcription of genes able to both promote survival, e.g., stimulating DNA repair, but also pro-apoptotic proteins such as PUMA and NOXA, members of the BCL-2 family of regulatory proteins [16].

## 3. The Apoptotic Pathways

Apoptosis is a well-characterised programme, first described in 1972 as being a morphologically distinct form of cell death [17]. Further studies of apoptosis revealed complex machinery of receptors, enzymes and organelles, all of which ensure correct and controlled execution of cell death. Critical to this execution is the group of cysteine-aspartic proteases known as Caspases, which, once activated, target numerous downstream cellular components to regulate their function [18]. All apoptotic caspases are initially present as inactive zymogens and must be activated to be catalytically active [19]. Activation of the initiator caspases occurs via either the intrinsic or extrinsic apoptosis pathways. Both pathways work together to ensure tight apoptotic homeostasis, removing compromised cells whilst limiting uncontrolled cell death [20].

The intrinsic or mitochondrial apoptotic pathway is triggered by stresses including oxidative stress, irradiation, DNA damage and treatment with cytotoxic drugs. Crucial in regulating the intrinsic pathway is the BCL-2 family of proteins, which can be divided into three groups based on their function: firstly, the multi-domain anti-apoptotic proteins, BCL-2, BCL-xL, BCL-w, MCL-1 and BFL-1/A1; secondly, the pro-apoptotic effector proteins BAK, BAX and BOK; and finally, the pro-apoptotic BH3-only proteins, BAD, BID, BIM, BMF, HRK, NOXA and PUMA [16]. The relative abundances and interactions that occur between the anti- and pro-apoptotic BCL-2 family proteins regulate the outer mitochondrial membrane potential and commitment of the cell to apoptotic cell death [21].

In response to internal damage, effector proteins BAK or BAX are activated at the outer mitochondrial membrane. Detailed structural and biochemical analysis has determined multiple steps required for both BAK and BAX activation [22,23], including conformational changes in the protein structure that are triggered by transient binding of a BH3-only protein (BID, BIM, etc.) or p53 [24]. The conformational changes enable BAK or BAX homo-dimers to form and subsequently higher-order oligomers, which cluster in the membrane disrupting it and causing the release of Cytochrome C and AIF (Figure 3) [25]. Activation of the effector proteins can be halted in two ways by the anti-apoptotic BCL-2 proteins. First, by directly binding to BAK and/or BAX, which prevents BH3-only proteins from interacting with them and triggering activation. Alternatively, the anti-apoptotic BCL-2 proteins can bind to and sequester the BH3-only proteins to prevent them from interacting with the BAK and/or BAX [16]. Therefore, it is the balance between anti-apoptotic and pro-apoptotic proteins in the cell that determines whether intrinsic apoptosis can proceed and thus play an essential role in cell fate decisions. Interestingly, many of the BCL-2 family proteins are transcriptional targets of p53 [26].

The extrinsic route to apoptosis, or the death receptor (DR) pathway, is mediated by transmembrane receptors that are members of the tumour necrosis factor (TNF) receptor protein superfamily [27]. These receptors contain a cytoplasmic death domain (DD) of around 80 amino acids, which is crucial in relaying extracellular signals from natural killer cells and macrophages, which release death ligands [28]. Each DR has its own corresponding death ligand, with the receptor/ligands driving apoptosis being FAS/FAS-L, TRAIL-R1/TRAIL and TRAIL-R2/TRAIL. Ligand binding induces receptor clustering in the membrane and induces a conformational change in the preassembled receptors, which leads to the recruitment of adaptor protein FADD via its DD [29]. FADD contains a death effector domains (DED) at the C-terminus, which facilitates the recruitment of procaspase-8 via DED interactions to form the death-inducing signalling complex (DISC) on the cytoplasmic side of the DR [30]. Biochemical analysis of the DISC revealed that FADD is present in sub-stoichiometric quantities relative to Caspase-8 [31,32], which led to the proposal, and later visualisation by CryoEM [33,34], of the DED chain model. The formation of a helical chain of Caspase-8 molecules, nucleated by FADD, ensures that the catalytic domains of two Caspase-8 molecules are correctly aligned so they can dimerise in an anti-parallel orientation forming the active site, auto catalyse, resulting in active Caspase-8 [33]. Once activated, Caspase-8, in turn, activates pro-caspase-3 via proteolytic cleavage to initiate the Caspase cascade [18].

p53 principally serves as an apoptotic regulator by modulating the expression of key proteins in both the intrinsic and extrinsic pathways, including BCL-2 family proteins, death receptors CD95 and Death Receptor 5 (DR5), APAF-1 and Caspases [35]. Additionally, p53 is able to modulate apoptosis via transcription-independent mechanisms, including an accumulation of p53 in the mitochondria that results in cytochrome c release and caspase activation [36].

Although initiation of apoptosis is often thought of as the point of no return, some cancers have found ways to reverse or halt the process and thus change cell fate, a process known as anastasis [37]. Several distinct mechanisms have been described to explain how cancer cells are able to escape apoptosis. Firstly, apoptosis can be reversed if the death-inducing stimulus is removed [38]. In the case of p53-induced apoptosis, p53 itself must be inactivated, and its apoptotic functions cease. This usually occurs via upregulation of MDM2, and therefore this could be a key marker of cells able to reverse the apoptotic programme [39]. Secondly, it has been reported that when p53 initiates apoptosis, it simultaneously activates the DNA repair programmes [40]. It is thought this acts as a fail-safe mechanism so that if the DNA damage cannot be repaired, the apoptosis process has already been initiated, and cell death can be executed rapidly, but it can also work as an opposing signal that counteracts the apoptotic pathway. Thirdly, incomplete apoptosis caused by the treatment of cancer cells with sub-optimal doses of drugs also triggers anastasis. For example, treatment of cancer cells with sub-lethal doses of first-generation BH3-mimetic compounds such as ABT-737 results in incomplete MOMP or minority MOMP [41]. These cells do not die and instead further contribute to more aggressive tumours. Additionally, long-term exposure to these agents caused mutations in the BCL-2 family proteins and consequently led to the acquisition of apoptotic resistance following prolonged exposure to these agents [42]. Therefore, in the context of cancer cells, anastasis can be a molecular mechanism utilised to change an apoptotic cell fate decision and enhance the tumourigenic phenotype in response to treatment with chemotherapeutic agents.

## 4. Non-Apoptotic Cell Death

In addition to cell death via the canonical apoptotic pathways, p53 has been reported to also be central to several forms of non-apoptotic cell death. One example of this is an alternative p53 driven cell death pathway first identified in experiments where caspase activity was inhibited using the inhibitor ZVAD.fmk [43]. In this study, the authors observed that p53 was able to drive a form of MOMP- and Caspase-3-independent cell death, although the precise molecular mechanism was not elucidated. Further studies into p53 dependent caspase-independent cell death revealed that p53 together with PARP-1 are able to drive a necrotic cell death in response to DNA damage and elevated ROS levels [44]. In addition to PARP-driven necrosis, p53 can drive programmed necrotic death by inducing cathepsin Q, which also cooperates with ROS to induce cell death [45]. These studies highlight the plasticity of cell fate decisions: if one cell death pathway is not available due to mutation or inhibition, p53 is able to drive the cell down an alternative pathway. This has far-reaching consequences for the treatment of cancer and suggests caution needs to be exercised when cell death pathways are targeted.

## 5. Cellular Senescence

Senescence was first observed in normal human cultured cells [46]. Human fibroblasts ceased to replicate after 50 cumulative population doublings (CPDs) in culture flasks, whilst cancer cells continued replicating indefinitely, leading to the hypothesis that cellular factors were behind this limitation in growth and that such arrested cells contribute to organismal ageing. In the 1970’s, the term Hayflick limit was coined to describe the number of times a human cell divides before ceasing replication [47].

A senescent cell may be identified as being in a state of irreversible cell cycle arrest, though metabolically active while displaying phenotypic changes. Other forms of growth arrest exist, such as quiescence and terminal differentiation, which are distinct from senescence in terms of the pathways triggered and the resultant phenotype. Senescence can be activated by numerous cell stressors, such as DNA damage, hypoxia, oncogene activation and nutrient deprivation, with most mitotic cells in the body, including cancer cells, able to undergo senescence [48,49]. If the stress is repairable or transient, it may lead to a temporary cell cycle arrest instead [49].

Senescence can occur due to perturbation of the balance between transcriptionally inert heterochromatin or the more open transcriptionally active form euchromatin. The extent to which each conformation exists depends largely on histone modifications such as acetylation or methylation. The nuclei of senescent cells contain areas of facultative heterochromatin known as senescence-associated heterochromatin foci (SAHF), which appear as punctate foci upon DAPI staining [50]. SAHF are also associated with the recruitment of RB to foci and associated with the repression of E2F target genes [50]. Additionally, a decrease in trimethylation of Lys27 on histone H3, due to a deficiency in the polycomb-repressive protein EZH2, can lead to senescence in human primary fibroblasts through upregulation of p16 [51]. Histone deacetylase inhibitors can also induce senescence mediated by p16 in human fibroblasts [52]. How chromatin perturbation exactly triggers senescence is not fully known, but it appears that chromatin reorganisation may be indicative of altered gene expression related to senescence.

Senescence can be caused by stresses such as ionising radiation due to an accumulation of double-strand breaks leading to the activation of ATM and a DDR (Figure 3) [53]. The use of radiotherapy to treat malignancies also induces senescence, and this has been termed therapy-induced senescence (TIS) [54]. Various cancer cells may undergo senescence in response to irradiation, but the resulting cell fate may depend on the genetic background and cell type. For example, cells that contain functional p53 may be more sensitive to stress and more likely to undergo senescence in response to radiation [55]. The ability to detect stresses and respond accordingly through altered gene expression appears to be necessary to develop a senescent-like phenotype [56].

Oncogene-induced senescence (OIS) occurs when normal cells undergo senescence in response to oncogene activation and acts as a tumour-suppressive mechanism that prevents benign lesions from progressing to malignant tumours [57]. RAS proteins transmit mitogenic signals from extracellular stimuli via G-protein coupled receptors and are frequently mutated in a number of cancers [58]. Primary human and rodent cells can enter G1 cell cycle arrest when oncogenic RAS is expressed with the accumulation of p53 and p16, thus inducing OIS [59]. Overexpression of the mutant BRAF oncogene is observed in many human melanoma patients, resulting in constitutively active protein and the development of melanocytic nevi, a benign skin tumour that is mostly comprised of senescent cells [57]. After driving hyperproliferation in melanocytes, mutated BRAF then stimulates p16 expression, leading to cell cycle arrest and senescence [60]. OIS induced by mutant RAS can be bypassed by abrogating the p16 pathway [59,60].

## 6. Changing Senescent Cell Fates with Therapeutic Purposes

Whilst the traditional definition of senescence implied that the cell cycle arrest had to be irreversible, some studies are now proposing that cell fate decisions can be changed. Modulating the key proteins involved in cell cycle arrest, such as p53 and p21, may allow cells to escape senescence. Whilst p53 levels in a non-stressed state are low, the pulsatile dynamics of p53 in stress may allow the cell options of quickly switching to sustained levels of p53 and arrest or a drop in levels and return to proliferation. This subtle balance may be therapeutically exploited in the right context to push tumour cells into senescence or potentially remove senescent cells from aged tissues. Additionally, there is growing evidence that the p53–p21 cell cycle arrest pathway is not simply unidirectional but that the dynamics of p21 may affect cell fate decisions following stress. For example, overexpressed p21 has been shown to render glioma cells resistant to chemotherapy-induced apoptosis by allowing the repair of damaged DNA [61]. Moreover, it has been shown that the early dynamics of p21 following stress may determine final cell fate [62]. Using a single-cell approach, three early p21 patterns have been described. Acute drug-induced p21 level leads to senescence. Similarly, a delayed p21 response also leads to senescence. On the other hand, an intermediate level between the two leads to proliferation. Counter-intuitively, cells fated to senescence expressed low levels of p21, whilst only a small proportion expressed high levels. This led to the concept of a “Goldilocks zone” of p21 expression that may ultimately foster proliferation and be crucial to determining cell fates. This suggests that simply increasing p53 or p21 levels during chemotherapy may result in tumour relapse; therefore, a more nuanced drug delivery approach to specific tumour types may be needed to achieve optimum expression levels and the desired cell fate.

Several studies have revealed functions for p21 aside from cell cycle arrest, which may contribute to determining cell fate. For instance, HCT116/p21^−/−^ cells display increased p53 levels without drug-induced DNA damage, which also correlates with an increased expression of p14^ARF^ that may promote p53 stability by binding to MDM2 [63]. These cells also displayed increased BAX levels, thereby altering the BAX/BCL-2 ratio, which, upon chemotherapeutic treatment, resulted in an increased sensitivity to apoptotic death. This demonstrates that p21 may contribute indirectly to apoptotic resistance by enhancing p53 stability in addition to causing cell cycle arrest. On the other hand, p21 may induce an apoptotic response instead, determined by intracellular reactive oxygen species levels and the specific cell sensitivity to oxidative stress [64]. All this highlights the importance of p21 in cell fate decisions and suggests that finding an optimum p21 level may allow a fine-tuned response following chemotherapy to shift the response to either proliferation, senescence or apoptosis, depending on the required outcome.

Mutant p53 can also cause normal cells to evade senescence [65]. Near senescent fibroblasts transfected with an alanine 143 p53 mutant managed to continue proliferating for a further 17 divisions, but this had no effect on the “younger” fibroblasts. This confirms the tumour-suppressive function of wild type p53 by imposing a cell cycle arrest and limiting tumour growth. Similarly, senescence can be reversed in MEFs by suppressing p53 expression using short hairpin RNA [66]. Interestingly, these newly proliferating cells exhibited reduced p21 levels but maintained high levels of p16, a central part of the p53-independent pathway to senescence. More recently, it has been reported that altered p53 dynamics may also allow cells to escape from cell cycle arrest. Cells that escaped senescence a week after irradiation, despite initially undergoing cell cycle arrest [67], displayed sharp switches between p21 and CDK2 expression, which suggests a double-negative feedback loop between the two that may reinforce escape from arrest.

Although therapeutic strategies that cause senescence have been developed over the last few decades, a growing number of studies have shown that TIS may not be a therapeutic endpoint as first thought. A small population of cells can evade growth arrest following a clinically relevant dose of chemotherapy and may contribute to a drug-resistant phenotype, aided by the cytokines and other factors secreted by the cells that have undergone TIS [68]. In particular, cancer cells deficient in both p16 and p53 have been shown to evade senescence after treatment and resume proliferation [69]. These escaped cells still expressed senescent markers and transcriptionally resembled their parent cells but overexpressed CDC2/CDK1, suggesting a possible pathway responsible for the change in cell fate. Other cells have also been shown to recover replicative potential following TIS [70].

This suggests that a tumour cell turned senescent may sometimes not be therapeutically relevant and may sometimes even contribute to resistance and relapse. One strategy to bypass this issue would be to first push tumour cells into senescence, followed by the selective killing of these cells using specific compounds (called senolytics). This “one-two punch therapy” approach was demonstrated in liver cancer cells containing p53 mutations induced to senesce by inhibiting CDC7 kinase, which was followed by sertraline treatment to induce apoptosis in these cells through mTOR inhibition [71]. Additionally, wogonin, a natural flavonoid, can induce p53 dependent senescence in T cell malignancies, which can then be treated with the senolytic Navitoclax (ABT-263) to further reduce cell viability [72]. These studies demonstrate that manipulating cell fate decisions can lead to the selective death of tumour cells that had initially been arrested, with potential therapeutic effects. Identifying modulators, interactors and other key players in the pathways leading to either survival or death should allow an improved response in specific therapeutic settings and an optimal response in patients.

## 7. Role of p53 in Cell Fate Decisions

At first glance, it would appear that senescence and apoptosis are two distinct cell fate pathways. However, many of the proteins induced are integral to both pathways. The ability of p53 to regulate both senescence and apoptosis makes it an important protein in tumour suppression, as confirmed by the fact that it is mutated in about 50% of all cancers [73]. The majority of these mutations are missense that still produces a full-length protein but is functionally inactive. The mutated p53 proteins can still interact with wild type p53 proteins, and the formation of these dimers and tetramers containing mutated forms of p53 inhibits their transcriptional activity [74].

p53 activation in response to oncogenic signalling is prevalent in pre-malignant and malignant cells [75]. OIS can increase the stability of p53 through p14^ARF^, which in turn is stabilised by oncogenic c-MYC transcriptionally upregulating USP10. p14^ARF^ stabilises p53 by sequestering MDM2 in the nucleolus and prevents MDM2 from binding and ubiquitinating p53 [76]. In addition to transcriptional regulation, p53 has been shown to induce ROS accumulation by controlling the expression of genes involved in the metabolism of ROS [77]. Conversely, p53 is also able to function as an antioxidant through expression of genes with an antioxidant function, such as TIGAR, GPX1, SOD2 and SESN1 [78]. p21 has also been implicated in ROS generation and the establishment of a senescent phenotype [64,79]. p21 is crucial in maintaining p53-dependent senescence by inhibiting cyclin E/CDK2 and progression from G1 to S phase [80]. p21-mediated cell cycle arrest is transient and can be abolished if the DNA damage is repaired. However, if the damage is slow to repair or faulty, then the ATM/ATR-p53–p21 signal would be sustained, and senescence ensues [49]. It has been proposed that p21 is able to lock a cell in a state of deep senescence through mitochondrial dysfunction and the generation of ROS, which continuously replenish short-lived DNA damage foci, resulting in a permanent senescent phenotype [81].

As mentioned above, p53 regulates apoptosis by both transcription-dependent and independent mechanisms, the latter requiring p53 to be transported out of the nucleus [82,83]. From the cytoplasm, p53 can then move to the mitochondria to interact with BCL-2 family proteins. The localisation of p53 is dependent on post-translational modification such as phosphorylation and acetylation. Specifically, phosphorylation of S392 or S15 has been shown to be important in apoptosis since this modification increases the ability of p53 to translocate to mitochondria, and this allows p53 to interact with proteins that can either inhibit or enhance apoptosis [84]. For instance, phosphorylation is especially important for PIN-1 mediated apoptosis. PIN-1 binds to the p53 phosphorylated forms of S33, S46, T81 and S315 and this binding causes the dissociation of MDM2, which reduces the proteasomal degradation of p53 [84]. Moreover, Pin-1 binding also prevents iASPP binding, which inhibits p53’s ability to induce apoptosis. Another important protein is LACE1, which is embedded in the inner mitochondrial membrane (IMM) and binds to p53, stabilising its translocation to the mitochondria [85]. Acetylation is also an important post-translational p53 modification since acetylation of Lys320, 373 and 382 releases BAX from Ku70, allowing it to be inserted into the MOM [86].

It has been observed that p53 binds to BAK and aids in oligomerisation. The binding of p53 to BAK is through the H2 helix, Loop 1 and Loop 3 within the DNA binding domain (DBD) of p53 to the N-terminal side of BAK, specifically the electropositive residues R280 (H2 helix), R248 (L3 loop) and K120 (L1 loop). These residues then interact with the electronegative residues E24, E25 and D160 on BAK [24]. The ability of p53 to bind to BAK opens the possibility that p53 binds to other BCL-2 family proteins, providing another mechanism that p53 regulates apoptosis independently from transcription. It was found that p53 can bind to BCL-xl through a similar electrostatic interaction as BAK. Interestingly, p53 has a higher affinity for BCL-xl than BAK [87], and p53 dimers can bind to BCL-xl with a higher affinity than p53 monomers [88]. The binding interface for the p53 dimer to BCL-xl is similar to the model for p53 monomers binding to BCL-xl, with numerous overlaps in important amino acids. p53 can also bind to MCL-1, but the binding site differs from BCL-xl and thus BAK. The corresponding negatively charged residues in the α1, α5 and α6 helices of BCL-xl that form interactions with the positively charged residues on p53 are either neutral or positively charged in MCL-1; thus, MCL-1 has an extremely low affinity for the DBD of p53. Despite this, MCL-1 has a much higher affinity for p53’s TADs than BCL-xl, particularly to TAD2. Due to the difference in p53 binding to MCL-1 and BCL-xl, it is possible that p53 can inhibit both, increasing the release of pro-apoptotic protein BAK [89].

## 8. ROS Involvement in Cell Fate Decisions

ROS have been known as chemical entities for over 100 years, and their importance in biological systems has been studied since the 1950s [90]. The reactive products of oxygen include hydrogen peroxide (H_2_O_2_), superoxide anion radical (O_2_^•^^−^), nitric oxide (NO), hydroxyl ion (OH^−^) and hydroxyl radical (OH^•^), the ions and radicals being highly reactive due to the presence of an unpaired electron. ROS are normally a by-product of mitochondrial oxidative phosphorylation in the electron transport chain (ETC) [91]. Not all electrons follow the ETC: some leak out and interact with oxygen to form O_2_^•^^−^ and may be converted to H_2_O_2_ by superoxide dismutases (SODs), which act as the first line of defence in controlling ROS levels [92,93]. Another form of redox defence is provided by glutathione, a tripeptide of glutamyl-cysteinyl-glycine, which is able to attract ROS by means of an exposed sulfhydryl group [94]. It has been demonstrated that the ratio of oxidised glutathione (GSSG) to reduced glutathione (GSH) may be an indicator of the level of cell stress [95]. ROS can cause damage to cellular components such as proteins and lipids, as well as DNA via base oxidation, leading to double-strand breaks and initiating a DDR [96]. As ROS are continuously produced, the cell maintains a balance between oxidant generation and removal. When that balance shifts to an increase in ROS, the cell may undergo oxidative stress, which, if persistent, may result in cell death or senescence (Figure 3) [97].

The connection between senescence and intracellular oxidants can easily be seen by the application of low concentrations of hydrogen peroxide to human diploid fibroblasts, which causes them to enter a senescent-like growth arrest [98]. Additionally, senescent cells have been shown to exhibit higher levels of ROS than normal cells [99]. Intracellular oxidants, rather than just causing indiscriminate cellular damage, may thus be seen as part of signalling pathways affecting cell fate decisions. This is supported by the fact that when sub-lethal doses of hydrogen peroxide are applied to fibroblasts, senescence is induced, whilst higher doses lead to apoptosis [100]. This is also accompanied by an increase in p53 and p21 levels, leading to G1 arrest [100]. ROS inhibitors can ameliorate both p53 mediated senescence and apoptosis, and the absence of the p53 target genes BAX or PUMA inhibited p53-mediated apoptosis and ROS increase [101]. This suggests that ROS levels play a key role in determining cell fate after p53 induction.

## 9. BCL-2 Family Proteins Role in Cell Fate Decisions

The anti-apoptotic BCL-2 family of proteins also plays important roles in regulating both apoptosis and senescence, influencing whether a cell undergoes cell death or survives. After oxidative stress, an increase in BCL-2 expression is observed during the growth arrest, but these levels decline during apoptosis [102]. Moreover, the anti-apoptotic BCL-2 family of proteins actually enable senescent cells to survive. There does not appear to be just one single mechanism by which BCL-2 is able to influence cell fates. For example, BCL-2 upregulates CDK inhibitor p27 and RB family member p130 [103,104,105] to inhibit cell cycle progression and determine whether a cell will undergo apoptosis or senescence [106]. There is increasing evidence to suggest that BCL-2 upregulation is required to initiate and maintain senescence, and this can occur both independently of p53 and p16 [107] or in a p53/p21-dependent manner [108]. The BCL-xL expression after damage also results in the induction of p53/p21-dependent senescence [109]. Conversely, upregulation of MCL-1 inhibits senescence, and its downregulation is required to enable cells to enter senescence [110]. Structural analysis revealed that the domains of MCL-1 required for its anti-apoptotic activity are distinct from those required for senescence [111].

## 10. Conclusions: The Complex World of Cell Fate Decisions

Cell fate decisions in response to stress were initially seen in a simplistic way as being mainly proportional to the damage exerted. Whether a cell that survived after the necessary repairs entered senescence or died seemed to depend on the magnitude of the insult. However, a more complex picture emerged over the years, particularly with the discovery of the many different functions of p53. For a while, p53 was seen as the decisive factor in these decisions until questions emerged about its ability to select on its own the most appropriate cell fate according to each situation. This gave rise to two antagonistic models. In the first one, p53 was proposed to induce either apoptotic or arrest genes in a selective way, mostly through differential post-translational modifications (p53 smart), while the alternative (p53 dumb) suggested that p53 would induce both sets of genes simultaneously and other factors would help determine which one would dominate [112].

Currently, the most accepted view is likely an amalgam of both hypotheses: cell fate decisions are still not completely understood but are seen as the result of the interaction of many different factors, probably with p53 still at the centre of an intricate network, but not alone at the commanding seat. The conjunction of these modulators is what eventually determines which p53 function eventually prevails [113]. Among these factors, p21, ROS and the levels of BCL-2 family proteins seem to be particularly relevant, having all been shown to be able to change cell fates after p53 induction. In this regard, cell fates are no longer considered fixed endpoints but dynamic states that can be shifted given the right combination of activation and/or inhibitions of cofactors. Thus, an apoptotic signal can be escaped, and a senescent arrest can be turned into cell death or simply bypassed. This great degree of flexibility allows cells to adapt to each particular circumstance but can also be exploited in diseases to overcome the different defence mechanisms.

Indeed, identifying the factors that modulate cell fates is particularly relevant in the context of cancer treatments in order to improve therapeutic outcomes. Senescence, once seen as a safe tumour-suppressor mechanism, is now a fate to be avoided since it is considered no longer irreversible or innocuous. Importantly, the accumulation of senescent cells after therapy has been shown to create a microenvironment in the tumour that enhances cancer cell survival and eventually fosters relapse [114]. Converting these senescent responses into apoptosis could thus greatly improve the outcome. p53 functions are likely to be at least partially inhibited in all cancers, and this may be mediated by over-expression of anti-apoptotic cofactors. These could be counterbalanced by increasing the presence of other factors that favour cell death, thus tipping the balance towards lethal cell fate decisions instead. Such interventions would increase the efficiency of therapies and reduce the chances of resistance and relapse.

More research is needed to fully understand how cell fate decisions are made and can be modulated. This could lead to a better understanding of an intricate network of biological events but also to improved therapies for many cancers. For this, it will be important to keep p53 centre stage but open the focus enough to include the many helpers that end up contributing to these processes.

## Figures and Tables

**Figure 1 cancers-14-02745-f001:**
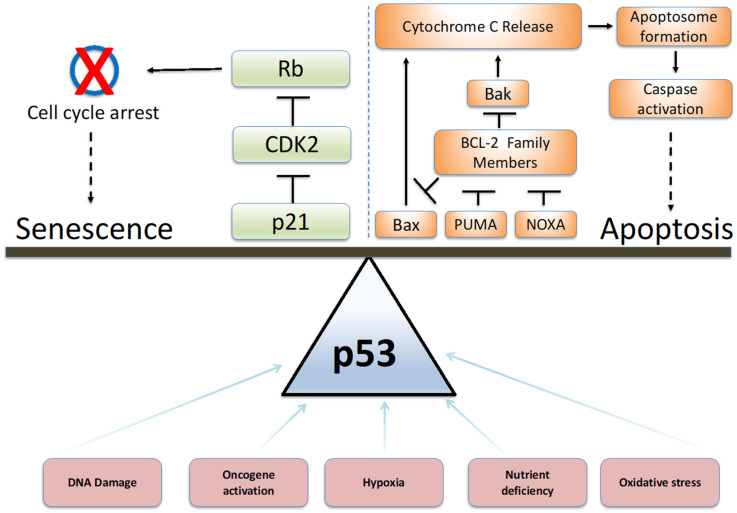
**p53 determines cell fate outcomes.** In response to different environmental stresses p53 dependent signalling is able to shift the balance between senescence and apoptosis.

**Figure 2 cancers-14-02745-f002:**
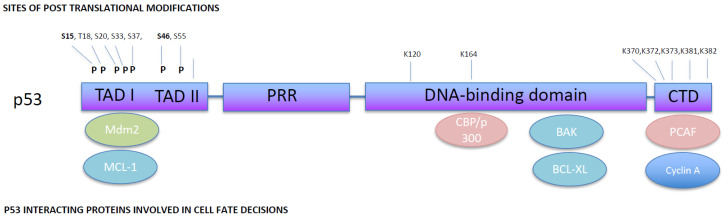
**Schematic representation of p53 domain structure and selected interactions with proteins involved in cell fate decisions.** p53 has two transcriptional activation domains (TAD I, residues 20–40 and TAD II, residues 40–60), a proline-rich region (PRR, residues 60–90), core DNA-binding domain (DNA-binding core, residues 100–300) and the C-terminal domain (residues 301–393). p53 has multiple post-translational modifications, including phosphorylation (P) or ubiquitination (Ub) on Lysine residues, affecting the cell fate outcome in response to p53 activation. Additionally, p53 interacts with specific proteins involved in cell fate, which also influences the cell fate outcome.

**Figure 3 cancers-14-02745-f003:**
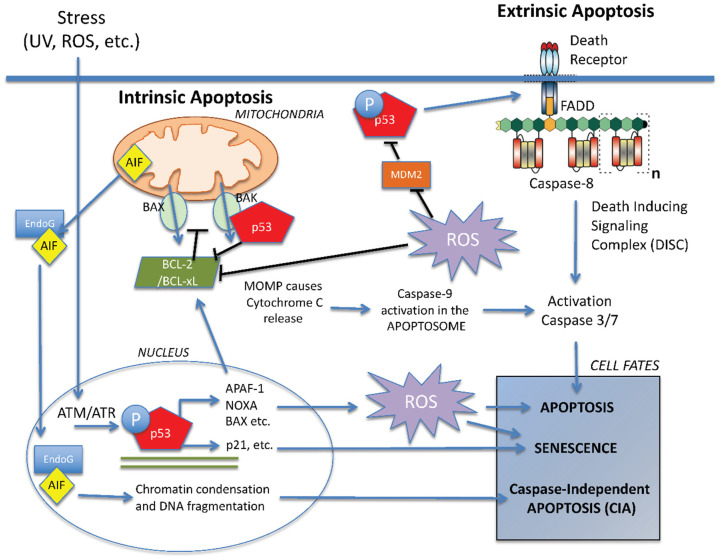
**Molecular mechanisms involved in cell fate decisions.** Summary of the main pathways involved in cell fate decisions after a cell has been subjected to stress. p53 interacts with multiple cellular signalling pathways to influence cell fate through both its transcriptional and non-transcriptional roles. ROS modulates these responses at different levels, having a key effect on senescence and the apoptotic pathways. BCL-2 proteins act as switch that can enhance or dampen cell death signals and thus favour arrest or death. It is the balance of all these factors, and others, that eventually determines cell fate after damage.

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
