# Peer review of "At the Crossroads of Life and Death: The Proteins That Influence Cell Fate Decisions"

_cancers, 2022, doi:10.3390/cancers14112745_

Round 1

Reviewer 1 Report

The resubmitted review paper by Vinesh Dhokia et al on “At the crossroads of life and death: The proteins that influence cell fate decisions” has been improved since the initial submission by the authors.

I feel the authors have answered all my queries raised before and I feel the review paper is suitable for publication, after collective reviewer decision. Thank you.

Okay. Yes, It is fine now. The manuscript is very well revised.it is suitable for acceptance after the editor's decision. 

Author Response

We thank the reviewer for their comments that allowed us to improved and strengthened our manuscript significantly. 

Reviewer 2 Report

The new version of the manuscript is improved 

Author Response

(The authors gave the same response as above.)

Reviewer 3 Report

     The manuscript has been extensively revised, most of the criticism and comments of the reviewers have been addressed. All these significantly improved the manuscript. Several typos, however remained in the text (even though all these were listed in my original review). These are:

line 32: ...p53 in cell fate.

Figure 1:  Oncogene activation, Nutrient deficiency, Oxidative stress

line 68: ...BRCA1...

For the three-letter abbreviation of human proteins all-capital-letter names should be used (BCL-2, FAS/FAS-L, RAS, CDC2/CDK1, c-MYC, CDK2, PIN-1)

line 121: ...(TNF) receptor protein superfamily.

There is no Chapter 4.

line 225: ...constitutively active protein...

line 244: ...p21 level leads...

line 261: ...reactive oxygen species levels...

line 374: ...oxidised glutathione (GSSG)...

Author Response

We thank the reviewer for their comments that allowed us to improved and strengthened our manuscript significantly, and confirm we have now corrected all the typos identified. 

This manuscript is a resubmission of an earlier submission. The following is a list of the peer review reports and author responses from that submission.